# Systemic Inflammation Is Associated with Pulmonary Hypertension in Isolated Giant Omphalocele: A Population-Based Study

**DOI:** 10.3390/healthcare10101998

**Published:** 2022-10-11

**Authors:** Baptiste Teillet, Mohamed Riadh Boukhris, Rony Sfeir, Sébastien Mur, Emeline Cailliau, Dyuti Sharma, Pascal Vaast, Laurent Storme, Kévin Le Duc

**Affiliations:** 1Department of Neonatology, Pôle Femme-Mère-Nouveau-Né, Hôpital Jeanne de Flandre, Centre Hospitalier Universitaire de Lille, F-59000 Lille, France; 2Center for Rare Disease Congenital Diaphragmatic Hernia, Jeanne de Flandre Hospital, Centre Hospitalier Universitaire de Lille, F-59000 Lille, France; 3Department of Pediatric Surgery, Jeanne de Flandre Hospital, Centre Hospitalier Universitaire de Lille, F-59000 Lille, France; 4ULR2694-METRICS: Évaluation des Technologies de Santé et des Pratiques Médicales, Axe Environnement Périnatal et Santé, Centre Hospitalier Universitaire de Lille, F-59000 Lille, France; 5Biostatistics Department, CHU Lille, F-59000 Lille, France; 6Department of Obstetrics, Jeanne de Flandre Hospital, Centre Hospitalier Universitaire de Lille, F-59000 Lille, France

**Keywords:** giant omphalocele (GO), newborn infant, delayed surgical closure, inflammatory syndrome, pulmonary hypertension

## Abstract

Our objective is to determine perinatal factors contributing to the development of pulmonary hypertension (PH) in patients with isolated giant omphaloceles (GO). All cases of omphaloceles that underwent prenatal and postnatal care at the University Hospital of Lille between 1996 and 2021 were reviewed. We included all infants with isolated GO, including at least a part of the liver, who were treated by delayed surgical closure. Prenatal and postnatal data were recorded and correlated with postnatal morbidities. We compared outcomes between a group of infants with GO who developed PH and infants with GO with no PH. We identified 120 infants with omphalocele. Fifty isolated GO cases fulfilled the inclusion criteria of our study. The incidence of PH was 30%. We highlighted a prolonged inflammatory state, defined as a CRP superior to 15 mg/L, platelets higher than 500 G/L, and white blood cells higher than 15 G/l for more than 14 days in patients who developed PH. This event occurred in 73% of patients with PH versus 21% of patients without PH (*p* < 0.05). Late-onset infection was not different between the two groups. We speculate that prolonged inflammatory syndrome promotes PH in infants with GO treated with delayed surgical closure.

## 1. Introduction

Omphalocele is a congenital disease with a defect of the abdominal wall closure that occurs in 3000 to 10,000 live births [1]. Due to the defect, part of the intra-abdominal organs moves out of the body during the intrauterine growth of the fetus. Giant omphalocele (GO) is a rare form of omphalocele. The most common description of GO refers to a large covered defect of the abdominal wall closure containing at least a part of the liver, even though there is no consensus on its definition [2]. Over recent decades, improvement in neonatal management increased the survival of patients presenting a GO, with a reported survival rate of 90% [3,4]. Surgical primary closure of the defect can be performed in small omphaloceles. Nevertheless, giant omphalocele surgical management is complex as the size of the defect makes it difficult to reach the edges of the defect contour (because of the gap, there is a lack of skin material). In addition, it is life-threatening to push back the sac content into the abdominal area as this maneuver would significantly increase the intra-abdominal pressure and therefore lead to cardiopulmonary failure [5]. Thus, delayed closure of the abdominal defect has been proposed as alternative care. In this way, the sac is topically treated to allow escharization and growth with little or no manipulation of the sac’s contents [3,6].

It is known that GO children experience ongoing medical and surgical morbidities. Gastroesophageal reflux is common in this population, as are impaired musculoskeletal development, neurodevelopmental delays, and nutritional disabilities [2,7,8,9,10]. However, the main challenge in neonatal care of GO is pulmonary hypertension (PH), which has been considered the main feature of giant omphalocele-associated adverse outcomes [11,12,13].

The pathophysiology of PH remains unclear. The main hypothesis implies a multifactorial mechanism, including pulmonary hypoplasia, vascular dystrophy, and aggression of assisted ventilation. Evidence exists in experimental PH that inflammation precedes vascular remodeling, highlighting that altered immunity may promote vascular disease [14,15]. Prenatal prognostic assessment in GO reveals that some neonatal outcomes are associated with pulmonary hypoplasia [4,16]. However, no significant association was found between prenatal lung hypoplasia measured by Magnetic Resonance Imaging (MRI) and PH. 

The objective of our study is to determine the perinatal factors associated with the development of pulmonary hypertension in GO patients. Our secondary objective is to describe mortality and morbidities in this population.

## 2. Population and Methods 

We designed a population-based study of patients presenting with omphalocele referred to the University Hospital of Lille between November 1996 and September 2021 for prenatal diagnosis. The national commission of information and liberty (CNIL) approved this study. 

We retrospectively reviewed all newborns with omphalocele in the Nord-Pas de Calais (North of France department) area admitted during the neonatal period in our center from 1996 to 2021. Among omphalocele cases, GO was defined as an omphalocele including at least a part of the liver [7]. All cases of prenatally diagnosed isolated giant omphalocele were referred to Lille University Hospital and included in a prospective cohort. Newborn infants with GO were included in the present study if they underwent at least one antenatal sonographic evaluation and if the pregnancy resulted in a live birth at the referral center. Exclusion criteria included GO patients with associated malformations or genetic syndromes, subjects with insufficient postnatal data, patients treated by primary closure of the septal defect, or patients with a documented plan for palliative care [16].

Data on patients’ prenatal follow-up, neonatal care, surgical repair, and respiratory support requirements and outcomes were recorded and analyzed.

### 2.1. Prenatal MRI and Echographic Acquisition 

Omphalocele diameter, omphalocele collar, and abdominal and cranial circumference were assessed from prenatal ultrasound studies. Pulmonary volumes were calculated from prenatal MRI (Ingenia, 3 Tesla in T2 sequence) using the Rypens et al. method [17]. Lung tissue was manually outlined in each slice of a single sequence, and volume was obtained by multiplying by slice thickness. Observed-to-Expected fetal lung volume was calculated by comparing that expected for gestational age from reference tables. The interpretation was made by radiologists who had experience with the technique to routinely assess lung volume in other diseases associated with pulmonary hypoplasia. Pulmonary hypoplasia was defined as an Observed-to-Expected pulmonary volume of less than 50%. 

### 2.2. Surgical Care

The main goal of the treatment consisted of closing the abdominal wall without inducing a life-threatening increase in intra-abdominal pressure [18]. The paint-and-wait treatment was chosen by our surgical team because this technique has previously shown a faster hospital discharge, initiation of full enteral feeding, and decreased incidences of late-onset infection [6].

The paint-and-wait procedure consisted of suspending the herniated sac by the umbilical cord immediately after birth. In addition, we applied a dressing on the omphalocele until the cutaneous epithelium was covered. In our center, we used fatty dressing (Vaseline Cooper©). This treatment allowed the closure of the abdomen by obtaining a cicatricle skin covering the medial ventral hernia. Definitive abdomen closure is proposed for children between 1-year-old and 2-years-old, mainly after walking achievement.

### 2.3. Postnatal Management

Neonatal medical records were studied, and neonatal outcomes were analyzed: gestational age at delivery, birth weight, APGAR score, blood gases, type of resuscitation, duration of mechanical ventilation, age at first and last closure procedure, duration of total parenteral and enteral feeding, timing to the first hospital discharge and major complications. Delayed surgical closure was used in all patients. 

Postnatal outcomes included the type of surgical repair, duration of ventilation and O2 supplementation, white blood cells (WBC) count, hemoglobin count, blood platelets count, C-reactive protein (CRP) concentrations, and gastroesophageal reflux disease with or without the need for surgery.

### 2.4. Late-Onset Infection 

We defined a late-onset infection as clinical sepsis associated with an elevation of CRP > 5 mg/L and proven by a positive bacterial culture (blood, urine, and cerebrospinal fluid). Every increase in CRP level was associated with multiple bacterial samplings (blood and urine). A late-onset infection could be septicemia, pyelonephritis, an infection of the central line, or pulmonary infection. 

### 2.5. Prolonged and Short Inflammatory Syndrome

Prolonged inflammation was defined as a CRP concentration above 20 mg/L for more than 15 days, a platelet count of more than 500 G/L for more than 15 days, and a WBC count of more than 15 G/L for more than 15 days, as opposed to the usual definition of acute inflammation which resolves in less than two weeks [19]. We defined brief inflammatory syndrome as a CRP level higher than 20 mg/L for less than ten days. We did not use average CRP because of the strong influence of extreme values, which made it difficult to interpret.

### 2.6. Primary Outcome

All patients underwent Doppler echocardiography by an experienced neonatologist or pediatric cardiologist in the first week after birth [20]. The following variables were measured in all patients: blood flow velocities in the left pulmonary artery and the ductus arteriosus (DA), pulmonary and/or tricuspid regurgitation gradient, and assessment of the shape of the interventricular septum. Pulmonary hypertension was defined as a combination of the following criteria:-Right-to-left or bidirectional flow in DA;-Flattening or paradoxical septum if DA is not present;-Mean blood flow velocity less than 0.25 m/s in pulmonary arteries;-Pulmonary or tricuspid regurgitation gradient above 50% of systemic systolic pressure if DA is not present;

Inhaled NO and/or Sildenafil were used in the case of PH. 

### 2.7. Statistical Analysis 

Categorical variables are expressed in terms of frequencies and percentages. Quantitative variables are expressed as means ± standard deviation (SD) in the case of normal distribution or medians (interquartile range (IQR)). The normality of distributions was checked graphically and by using the Shapiro–Wilk test.

Patients with PH were compared to patients without PH using the Chi-square test (or Fisher’s exact test in case of expected value < 5) for categorical variables and the Mann–Whitney U test for quantitative variables.

Statistical testing was conducted at the two-tailed α-level of 0.05. Data were analyzed using the SAS software version 9.4 (SAS Institute, Cary, NC, USA).

## 3. Results

### 3.1. Population

One hundred and twenty omphaloceles were assessed for eligibility between November 1996 and September 2021. As we can see in Figure 1, the final studied population included 50 patients with isolated giant omphaloceles. Antenatal and neonatal characteristics are described in Table 1 and Table 2, respectively. Except for a slight difference in Apgar score at 1 min, there were no other significant differences in neonatal characteristics between the two groups. 

### 3.2. Fetal Echographic and MRI Assessment 

The median omphalocele’s collar circumference in the second and third trimesters of gestation was 25 mm and 39 mm, respectively. The median gestational age at fetal MRI evaluation was 32 weeks. The mean pulmonary volume was 37.9 +/− 9.7 mL. Among 26 giant omphalocele cases who underwent fetal MRI, 13 showed a pulmonary volume of less than fifty percent. 

### 3.3. Outcomes

The mean duration of hospitalization was 62 days. Overall survival until discharge was 96%. Fifteen patients (30%) developed pulmonary arterial hypertension within the first month of hospitalization. Two children died because of PH at 132 and 180 days of age. Nitrogen monoxide was used in 24% of cases, and Sildenafil was used in 14%. Among GO neonates, 20% were intubated on the first day of life. The mean duration of invasive ventilation was five days. Three (6%) of the GO neonates required tracheostomy placement. The mean duration of parenteral nutrition was 28 days. Ten patients (20%) needed gastrostomy tube placement. Furthermore, 54% of GO patients needed medication such as omeprazole for gastroesophageal reflux disease, and 10% underwent a Nissen fundoplication. All of our patients were treated with delayed surgical closure; the median age of surgical closure was 360 days of life.

There was no significant difference between the two populations of GO (except at 1 min Apgar), although a trend was identified toward infants with PH being smaller in weight and gestational age. As we can see in Table 1, patients with PH showed no difference in terms of pulmonary hypoplasia according to Rypens et al., with pulmonary hypoplasia occurring in 20% of patients in the PH group and 17% of patients in the no PH group (*p* = 0.64). There were no significant differences in other measures, such as the ratio of omphalocele circumference and abdominal circumference or the ratio of omphalocele circumference and cranial circumference. As we can see in Table 3, two deaths occurred in the PH group (13%), but none occurred in the group without PH. We observed an inflammatory state, defined as an increase in CRP level of more than 20 mg/L over more than 15 days during the first month of life, occurring in 73% of patients with PH versus 21% of patients without PH (*p* < 0.05). Furthermore, brief inflammation duration is not associated with PH. As shown in Figure 2, there was no difference between the two groups with proven secondary infection, showing that this inflammatory state was not related to bacterial infection.

## 4. Discussion 

The aim of this study is to determine the prenatal and neonatal factors associated with the development of PH in patients with isolated giant omphaloceles, considering that PH plays a key role in morbidity and mortality in this population. We designed a retrospective population-based study where all patients with giant omphaloceles who were born in our region were included from 1996 to 2021. Several limitations must be acknowledged. Because of its retrospective design, we cannot rule out that some patients may not have been included in the study. However, it is unlikely, as special care has been taken to ensure the completeness of the reports by matching different databases. Although the rate of late-onset infection was not significantly different between the two groups, we cannot exclude that the difference may exist in a larger study, particularly due to the extension of the duration of invasive management in infants with PH. Furthermore, a single observer and a lack of MRI lung evaluation limited the interpretation of pulmonary hypoplasia. 

However, this study is one of the largest to examine the prenatal and postnatal outcomes of giant omphaloceles. The survival rate is high (96%) in our study and consistent with the literature. However, one-third of giant omphalocele cases in our population developed PH, and our results indicate that a prolonged inflammatory syndrome is associated with an increased risk of PH. GO newborns with PH have a lower survival rate and higher respiratory and digestive morbidities. 

In general, the omphalocele sac can exhibit considerable heterogeneity regarding size or contained viscera, and additionally, the presence of other malformations or chromosomal abnormalities significantly influences the prognosis of the affected newborns. For instance, isolated small omphaloceles are known to have a favorable prognosis. Conversely, isolated GO is associated with more respiratory failure, delayed full enteral feeding, a higher need for GERD medication or Nissen surgery, and neurodevelopmental delay [20,21,22]. A high incidence of digestive morbidities was found in our cohort. Furthermore, Partridge et al. showed that 37 percent of GO patients developed PH, which is consistent with our data. PH is known to be associated with longer NICU length of stay, longer mechanical ventilation, and almost 15% mortality [20]. This is in alignment with our study, which showed a significant excess of mortality in the GO with PH group. 

Vascular tonicity is frequently abnormal in pulmonary hypoplasia and induces a significant increase in mortality and poor long-term outcomes in patients with lung maldevelopment, such as congenital diaphragmatic hernia (CDH) [23,24]. In CDH, a rise in pulmonary vessel reactivity has also been associated with pulmonary hypoplasia, including decreased response to vasodilator stimuli [25]. However, Danzer et al. failed to show a link between lung volume and PH, probably because of the more complex pathophysiology in GO [16]. In our study, the prenatal markers of pulmonary hypoplasia or the size of the GO were not associated with PH. Although studies suggested that a narrow thorax and omphalocele on abdominal circumference ratio are associated with PH [26], they failed to understand the whole mechanism underlying PH-related GO. Indeed, PH tends to worsen in the first weeks after birth, suggesting postnatal mechanisms. Our results indicate that some infants with giant omphalocele exhibit prolonged inflammatory syndrome that is mostly independent of infection. The data provide evidence that PH’s development in GO is associated with sustained inflammation. To the best of our knowledge, the present study is the first to highlight that inflammation is part of PH development in GO. 

However, it is described in the literature that the inflammation process is linked to altered vascular cell metabolism. Correlation of the average perivascular inflammation score with vascular thickness with respect to mean pulmonary arterial pressure has been reported [15]. The fact that inflammation precedes vascular remodeling in experimental PH suggests that altered immunity is a cause of vascular disease [14]. Biomolecular studies show that activated CD8+ T cells contribute to the pathogenesis of PH through TNF-alpha activation. Several changes through this mechanism result in proliferation within the pulmonary artery wall, a hallmark of pulmonary arterial hypertension [27]. IL-6 overexpression in a mouse model induced the development of PH through the induction of FGF2 and the activation of the transcription factor KLF5 [28]. In addition, it has been proved in both experimental and clinical studies that neutrophil elastase can influence pathogenesis [29]. Indeed, in the mouse model, elastase inhibitor elafin repressed the development and progression of PH [29,30,31]. 

At our center, patients with evidence of elevated pulmonary pressures were assessed by echocardiography approximately once a week. Taking into account the one-third rate of PH in giant omphalocele, screening for PH should be performed within the first week of life in all patients, with echocardiography performed at regular intervals. The association of inflammatory syndrome and PH associated with GO is described for the first time. Mechanisms are not fully delineated and could be multifactorial. We hypothesize that the inflammatory syndrome emerges from a primary lesion of the liver or is related to delayed abdominal closure and skin inflammation. Currently, we use a delayed closure of the abdominal defect that could be responsible for significant inflammation around the peritoneal sac linked to local and chronic infection between the skin and the peritoneal sac. Several studies have shown interest in treating inflammation associated with PH [32]. However, further studies are required to evaluate whether inflammation management may improve the prognosis and management of short- and long-term complications.

## 5. Conclusions 

Our series highlights that pulmonary hypertension may worsen the outcome of giant omphalocele patients. We identified that prolonged inflammatory syndrome is associated with PH development. Further studies focusing on inflammation and altered immunity in giant omphalocele treated by delayed surgical closure may promote a new management strategy to prevent GO-related morbidity. 

## Figures and Tables

**Figure 1 healthcare-10-01998-f001:**
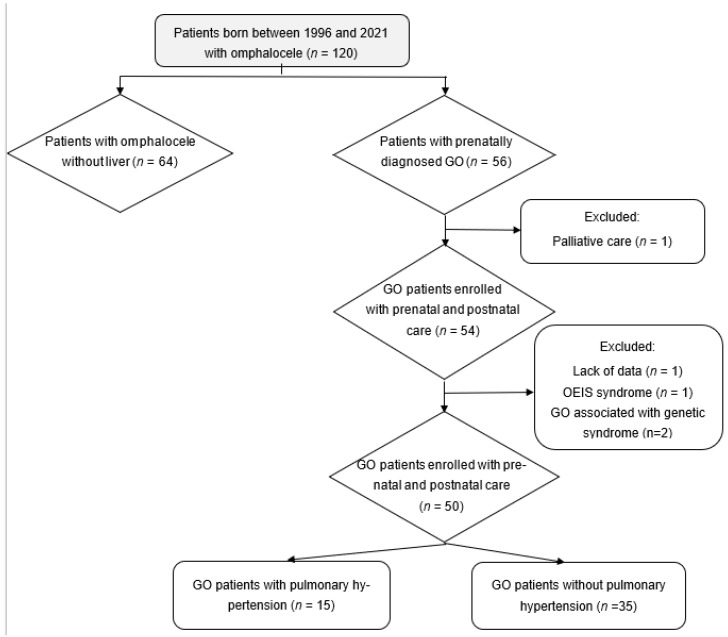
Flow chart of the study population. Abbreviation: GO, giant omphalocele, OEIS, omphalocele-cloacal exstrophy syndrome.

**Figure 2 healthcare-10-01998-f002:**
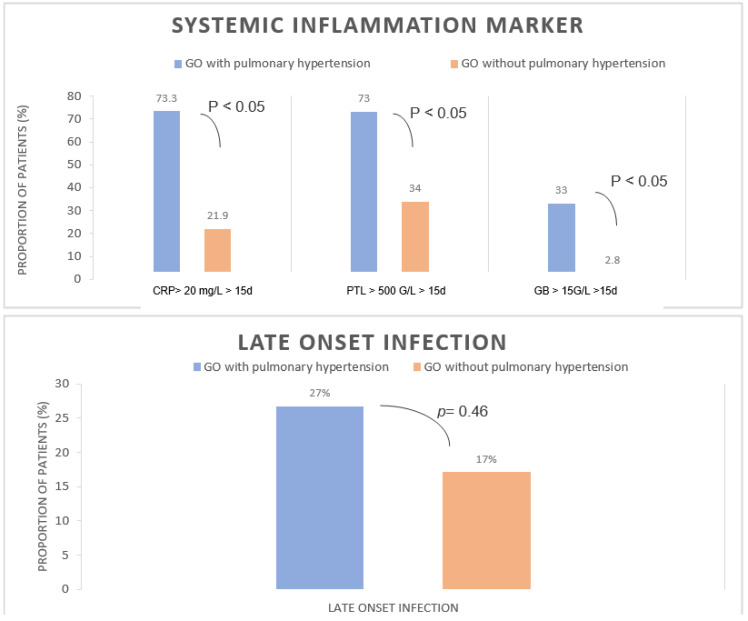
Comparison of systemic inflammation markers in GO with and without PH onset. Comparison of late-onset infection between GO with PH and GO without PH. Abbreviations: CRP, protein C-reactive; WBC, white blood cell; d, days; GO, giant omphalocele; PLT, platelet.

**Table 1 healthcare-10-01998-t001:** Antenatal characteristics.

	GO with PH (n = 15)	GO without PH (n = 35)	*p*-Value
Pulmonary hypoplasia (Rypens) (n = 28)	3/7 (43)	6/21 (27)	0.64
Ratio CO/CA at T2 (n = 10)	1.01 [0.97–1.05]	0.82 [0.74–0.93]	
Ratio CO/CA at T3 (n = 8)	0.95 [0.89–1.02]	0.72 [0.65–0.75]	
Ratio CO/CC at T2 (n = 10)	0.67 [0.66–0.80]	0.67 [0.58–0.71]	
Ratio CO/CC at T3 (n = 10)	0.64 [0.63–0.73]	0.64 [0.58–0.67]	
GO collar at T2 mm (n = 30)	25 [22–30]	24 [20–30]	0.76
GO collar at T3 mm (n = 24)	39 [37–45]	47 [29–52]	0.83
GO collar at MRI evaluation mm (n = 18)	38 [35–44]	40 [30–50]	0.93

Values are expressed as numbers/total numbers (percentage) or medians (interquartile range). Abbreviations: d, days; GO, giant omphalocele; CO, omphalocele circumference; CA, abdominal circumference; T2, second trimester; T3, Third trimester.

**Table 2 healthcare-10-01998-t002:** Neonatal characteristics.

Variable	GO with PH (n = 15)	GO without PH (n = 35)	*p*-Value
Male, n (%)	5 (33)	17 (48)	0.5
Birth weight (g), mean ± SD	2586 ± 792	2838 ± 1000	0.44
GA at delivery (wks), mean ± SD	36 ± 3	37 ± 2	0.1
Apgar at 1 min, median [IQR]	9 [4–10]	10 [6–10]	<0.05
Apgar at 5 min, median [IQR]	10 [8–10]	10 [8–10]	0.2
Venous umbilical cord pH^1^, mean ± SD	7.32 ± 0.04	7.33 ± 0.06	0.6

Values are expressed as numbers (percentage) unless otherwise stated abbreviation: GA, gestational; IQR: interquartile range; SD: standard deviation.

**Table 3 healthcare-10-01998-t003:** Outcome.

Variable	GO with PH (n = 15)	GO without PH (n = 35)	*p*-Value
DOL at final abdominal closure (d)	425 [13–790]	296 [1–1125]	0.51
Duration of parenteral feeding (d)	43 [5–381]	9 [0–40]	0.09
Enteral nutrition DOL at initial feedings (d) DOL on full goal volume feedings (d)	5.5 [0–30] 43 [5–375]	1 [0–15] 13 [0–40]	0.16 <0.05
Need of gastrostomy tube, n (%)	7 (46)	3 (8)	<0.05
Length of supplemental O2 > 30% (d)	8 [0–60]	0 [0–5]	<0.05
Need for GERD medication, n (%)	13 (86)	15 (43)	<0.05
Need for tracheostomy, n (%)	2 (13)	1 (2.8)	0.2
Duration of mechanical ventilation (d)	15 [0–56]	0 [0–8]	<0.05
Mechanical ventilation > 14 d, n (%)	9 (60)	0 (0)	<0.05
Death, n (%)	2 (13)	0 (0)	<0.05

Values are expressed as numbers (percentage) or median [IQR]. Abbreviation: d, days; DOL, day of life; g, gram; wks., weeks; GO, giant omphaloceles; GERD, gastroesophageal reflux disease; IQR: interquartile range; SD: standard deviation.

## Data Availability

The study data are available upon request.

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
