# Peer review of "Systemic Inflammation Is Associated with Pulmonary Hypertension in Isolated Giant Omphalocele: A Population-Based Study"

_healthcare, 2022, doi:10.3390/healthcare10101998_

Round 1

Reviewer 1 Report

The submitted article by Teillet et al investigates the perinatal factors contributing to the development of pulmonary hypertension in patients with isolated giant omphaloceles. The paper is well-written and novel in its conclusions. However, the paper would be further strengthened by addressing the following major and minor issues:

·       Abstract: line 28: Instead of ‘We compared outcomes between a group of infants with GO who developed PH and the other uninjured’, I would suggest writing ‘We compared outcomes between a group of infants with GO who developed PH and infants with GO with no PH.’

·       Page 3: line 99-100: ‘The main goal of the treatment consists in closing the abdominal wall without inducing life threatening increasing intraabdominal pressure’. Please reword it as ‘The main goal of the treatment consists of closing the abdominal wall without inducing a life-threatening increase in intraabdominal pressure.’

·       Page 3: line 104-105: ‘The paint-and-wait procedure consists in suspending by the umbilical cord the herniated sac immediately after birth.’ Please rewrite it as ‘The paint-and-wait procedure consists of suspending the herniated sac by the umbilical cord immediately after birth.

·       You used treatment with inhaled nitric oxide (iNO) or sildenafil at any time point during the infant's hospitalization as one of the criteria to define PH. Were there any infants that were treated with these medications because of clinical deterioration and/or increasing FiO2 needs who did not have evidence of PH on echo?

·       Page 5: Fig. 1: Please address the following:

1)     Many words are out of the boxes in the figure.

2)     Line 178 seems incomplete: ‘GO associated with generic’.

3)     Line 183: Please complete the sentence with a value for n. ‘GO patients without pulmonary hypertension (n)’

·       Please mention that there was a trend toward infants being smaller in weight and in GA in the group with PH as per table 2 as being SGA has been associated with PH.

·       Page 7: table 3: Please correct the heading ‘GO without PH’ which is written twice.

·       As per Fig.2, you showed that 27% of infants with GO with PH had late onset infection compared to 17% in GO without PH. Although, the difference was not statistically significant in your study with n=50, do you think this could become significant in a larger study and may explain the difference in the rate of PH in this group?

Reviewer 2 Report

This is retrospective observational study looking at perinatal factors associated with risk of pulmonary hypertension (PH) in neonates with prenatal diagnosis of giant omphalocele (GO). The study concludes that ‘prolonged inflammatory state’ was associated with increased risk of PH diagnosis in this population. The article is easy to follow. The following concerns need to be addressed by the authors. 

1)    Line 64: this sentence needs to be paraphrased – this is an observational study and can only comment on ‘association’ and not causation. 

2)    Line 128: How did the authors come up with the definition of prolonged and short inflammatory syndrome – can they authors cite any references for the duration or inflammatory marker cutoffs that they used for these definitions? 

3)    For clarity, I would suggest the authors present the average CRP values during hospitalization for patients in the two groups for comparison.

4)    Line 212-220: This section needs to be re-written – there seems to be multiple errors in the text with wrong table callouts or no mention of the data tables at all. Some of the numbers seems off when compared to the table however not sure if the numbers denote something else. If that is the case, the authors need to clearly delineate these things for the reader. 

5)    A paragraph denoting weakness and caveats of the study should be included in the discussion sections. 

6)    The language in the conclusion needs to be softened – again, this is an association and not causation and that should be clearly explained. 

Round 2

Reviewer 1 Report

Thank you for your reply and for addressing all the questions. I have only 1 following question/concern:

1. You mentioned that all the infants who were labeled as having PH were based on echo results as per the local protocol but you also mentioned that there were many infants who did not have echo reports available. Can you please explain this? Is it due to the unavailability of echo reports that were done or because the echo was not done? Is it possible that there may be infants who were treated as PH due to clinical suspicion with no echo evidence and did not have PH that were included in the group who had PH in the study? If you think this is possible please mention this in the limitations as it may change how to interpret the results.

Reviewer 2 Report

Congratulations to Teillet et al. for an interesting manuscript. I am satisfied with the revision and the author's comments.

Author Response

Baptiste Teillet

Department of Neonatology

University Hospital Jeanne de Flandre,

CHRU of Lille, F-59000

E-Mail:           [email protected]

Lille, June 22th 2022

Dear reviewer,

Thank you very much, once again, for your help in the improvements of our study.

Yours sincerely

Teillet Baptiste